# Effectiveness of COVID-19 Vaccination with mRNA Vaccines for Patients with Cirrhosis in Hungary: Multicentre Matched Cohort Study

**DOI:** 10.3390/vaccines11010050

**Published:** 2022-12-26

**Authors:** Bálint Drácz, Veronika Müller, István Takács, Krisztina Hagymási, Elek Dinya, Pál Miheller, Attila Szijártó, Klára Werling

**Affiliations:** 1Department of Surgery, Transplantation and Gastroenterology, Semmelweis University, 1083 Budapest, Hungary; 2Department of Pulmonology, Semmelweis University, 1083 Budapest, Hungary; 3Department of Internal Medicine and Oncology, Semmelweis University, 1083 Budapest, Hungary; 4Digital Health Department, Semmelweis University, 1083 Budapest, Hungary

**Keywords:** liver cirrhosis, COVID-19 vaccination, mRNA vaccines, vaccine effectiveness

## Abstract

Patients with cirrhosis are vulnerable to hepatic decompensation events and death following COVID-19 infection. Therefore, primary vaccination with COVID-19 vaccines is fundamental to reducing the risk of COVID-19 related deaths in patients with cirrhosis. However, limited data are available about the effectiveness of mRNA vaccines compared to other vaccines. The aim of our study was to investigate the efficacy of mRNA vaccines versus other vaccines in cirrhosis. In this retrospective study, we compared clinical characteristics and vaccine effectiveness of 399 COVID-19 patients without cirrhosis (GROUP A) to 52 COVID-19 patients with cirrhosis (GROUP B). 54 hospitalised cirrhosis controls without COVID-19 (GROUP C) were randomly sampled 1:1 and matched by gender and age. Of the cirrhosis cases, we found no difference (*p* = 0.76) in mortality rates in controls without COVID-19 (11.8%) compared to those with COVID-19 (9.6%). However, COVID-19 patients with cirrhosis were associated with higher rates of worsening hepatic encephalopathy, ascites and esophageal varices. Patients with cirrhosis receiving mRNA vaccines had significantly better survival rates compared to viral vector or inactivated vaccines. Primary vaccination with the BNT162b2 vaccine was the most effective in preventing acute hepatic decompensating events, COVID-19 infection requiring hospital admission and in-hospital mortality.

## 1. Introduction

Severe acute respiratory syndrome (SARS-CoV-2) causing COVID-19 disease has emerged as a pandemic around the globe. Although comorbid conditions such as diabetes, hypertension, chronic lung diseases (particularly COPD), chronic kidney disease and cardiovascular disease have been reported as major risk factors for COVID-19 mortality, the prognostic value of chronic liver disease (in particular liver cirrhosis) is still undefined [1,2]. However, patients with cirrhosis following SARS-CoV-2 infection proved to be at a higher risk for poor outcomes. Registry data showed that overall mortality in patients with cirrhosis following COVID-19 infection ranged from 16% to 42% and poor outcome increased stepwise with the severity of cirrhosis [2,3,4,5]. Furthermore, patients with decompensated cirrhosis following SARS-CoV-2 infection were more susceptible to Intensive Care Unit (ICU) admission, renal replacement therapy and invasive mechanical ventilation compared to those without chronic liver disease (CLD) [6]. Acute-on-chronic liver failure (ACLF), which is a life-threatening syndrome, occurred in 12-50% of decompensating patients in COVID-19 infection [2,4,7]. Despite respiratory failure being the leading cause (71%) in COVID-19, liver-related manifestations (19%) are also common [2]. Nevertheless lung injury and hepatic dysfunction are tied by multiple links such as an altered gut–lung axis, gut dysbiosis, cirrhosis-associated immune dysfunction (CAID) and pulmonary complications due to ascites [8]. Primary vaccination is the most effective tool for preventing severe SARS-CoV-2 infection and is especially recommended for vulnerable individuals including patients with liver cirrhosis [9,10,11]. Notably, patients with cirrhosis were reported to display an accelerated decline in antibody titres compared to healthy individuals [12,13]. In Hungary since 15 January, 2021 five different COVID-19 vaccines have been commonly used: two mRNA-based vaccines (BNT162b2-Pfizer-BioNTech, mRNA-1273-Moderna), two vector vaccines (AZD1222-Astra Zeneca, Gam-COVID-Vac-Sputnik V) and one inactivated vaccine (HB02-Sinopharm) [14]. To date, there are limited data about the efficacy of different SARS-CoV-2 vaccine platforms (mRNA; viral vector; inactivated) in patients with cirrhosis. 

Therefore, we aimed to investigate the impact of primary vaccination with different SARS-CoV-2 vaccines on patients with cirrhosis. Furthermore, the aim of our study was to evaluate the effectiveness of mRNA vaccines, as compared to other COVID-19 vaccines, in cirrhosis.

## 2. Materials and Methods

### 2.1. Patient Cohorts 

We performed a retrospective multicentre study by reviewing data of COVID-19 patients hospitalized in several centres of Semmelweis University between March 2020 and May 2022. Diagnosis of SARS-CoV-2 required a positive reverse transcription-polymerase chain reaction (RT-PCR, SEQONCE qPCR Multi Kit, IVD) test on nasopharyngeal swabs based on the protocol by the World Health Organization [15]. Furthermore, High-Resolution Computer Tomography (HRCT, Phillips Incisive 128) was used on admission for the initial diagnosis of COVID-19 pneumonia. A total of 6394 COVID-19 patients were recruited for our analysis. Inclusion and exclusion criteria are listed in Table 1. We included 451 COVID-19 patients aged ≥ 18 years with elevated liver transaminases (>40 U/L) on hospital admission and/or underlying chronic liver disease (CLD) in the medical history. Patients with CLD were identified as cirrhosis cases. Diagnosis of liver cirrhosis was previously confirmed with liver biopsy, liver elastography, clinical presentation of portal hypertension (e.g., gastrointestinal varices on endoscopy), severity scoring systems (e.g., Child–Turcette–Pugh score) and radiomorphological findings (e.g., ascites, liver surface nodularity). Of 451 COVID-19 patients, we selected 399 COVID-19 patients without liver cirrhosis (GROUP A) and 52 COVID-19 patients with cirrhosis (GROUP B) and compared them with each other according to patient characteristics and vaccination status on admission. Thus, GROUP B patients were matched with 54 cirrhosis patients without COVID-19 infection (GROUP C) with respect to age and gender (Figure 1). The matching ratio was very close to 1:1. The representation of patients in cirrhosis groups was proportional. The cirrhosis controls without COVID-19 cohort included all laboratory-confirmed SARS-CoV-2 negative, adult patients who were previously diagnosed with liver cirrhosis and were admitted to the participating centres due to deterioration in cirrhosis stage between March 2020 and May 2022. We excluded patients with ambiguous diagnosis of liver cirrhosis or SARS-CoV-2 status not tested with RT-PCR. Furthermore, COVID-19 patients without liver injury on admission were excluded from the study. 

Based on our national protocol, indication for oxygen administration was arterial partial pressure of oxygen (PaO_2_) <60 mm Hg or peripheral oxygen saturation (SpO_2_) <90% at room air [16]. The initial strategy was a low-flow system up to 2 L/min via nasal cannulae and was increased over 6 to 15 L/min with reservoir facemask. For those requiring low-flow oxygen, a 5-day-long remdesivir regimen and dexamethasone 8–16 mg oral/8–16 iv were administered [16]. Patients with a persistent need for hig- flow oxygen (>60 L/min) or showing rapid progression over hours, hemodynamic instability or multiorgan failure needed mechanical ventilation.

Our study protocol was approved by the Scientific and Research Ethics Committee of the Medical Research Council of Hungary (IV/7946-1/2021/EKU, Budapest, 14 October 2021). It conforms to the ethical norms and standards in the Declaration of Helsinki.

### 2.2. Data Collection 

This retrospective study was conducted using data from electronic medical records. All data recorded were epidemiological and clinical characteristics, comorbid conditions, imaging features and COVID-19 therapy. The patients were followed until discharge or death. Criteria for hospital discharge of COVID-19 patients were as follows: (1) resolution of fever for >48 h without antipyretics, (2) oxygen saturation ≥ 94%, (3) no signs of increased work of breathing or respiratory distress, (4) improvement in signs and symptoms of illness (cough, shortness of breath and oxygen requirement) and (5) two negative RT-PCR tests in a row, at least 24 h apart. Admitted COVID-19 patients with underlying CLD were grouped by the presence of cirrhosis. Data about the most common causes of liver cirrhosis were retrieved from patient clinical reports. 

### 2.3. Liver Cirrhosis Severity

The classification system used to grade the severity of liver cirrhosis was the modified Child–Turcotte–Pugh Score (CTP). To estimate the prognosis of patients with cirrhosis, we registered parameters based on serum concentrations of total bilirubin and albumin, international normalized ratio (INR), degree of ascites and degree of hepatic encephalopathy on hospital admission and followed these parameters until hospital discharge or death. A CTP score of 5 to 6 is considered CTP class A (well-compensated disease), 7 to 9 is CTP class B (significant functional compromise) and 10 to 15 is CTP class C (decompensated cirrhosis) [17]. Cirrhosis patients were divided into 3 severity groups: compensated, decompensated and acute-on-chronic liver failure (ACLF). Decompensated cirrhosis was characterised by ascites, hepatic encephalopathy or variceal haemorrhage. The evaluation of ascites was based on the volume of abdominal fluid: mild ascites detected by ultrasound; moderate ascites described by proportional abdominal distension; large ascites with compelling abdominal extension [18]. The severity of hepatic encephalopathy was classified according to the West Haven Criteria [19]. Cirrhosis patients with acute decompensation were assessed for prognosis based on the European Association for the Study of the Liver Chronic Liver Failure (CLIF) consortium definition [20]. Acute-on-chronic liver failure (ACLF) was defined in accordance with the European Association for the Study of the Liver Chronic Liver Failure (CLIF) consortium definition and by the North American Consortium for the Study of End-Stage Liver Disease (NACSELD) [21,22].

### 2.4. COVID-19 Vaccines 

In Hungary, Pfizer-BioNTech was the first vaccine to be used in cirrhosis. A primary vaccination campaign using mRNA vaccines (Pfizer-BioNTech and Moderna) started on 15 January 2021. During the third wave in March and April 2021, five different vaccines were widely used in cirrhosis: two mRNA-based vaccines (BNT162b2-Pfizer-BioNTech and mRNA-1273-Moderna), two vector vaccines (AZD1222-Astra Zeneca, Gam-COVID-Vac-Sputnik V) and one inactivated vaccine (HB02-Sinopharm) [14]. Patients included in our study were unvaccinated or had undergone primary immunization (were 14 days after receiving two doses of Pfizer-BioNTech or Moderna, or one dose of Pfizer-BioNTech and one dose of Moderna, or two doses of Sputnik or Sinopharm), or had already received booster vaccination following the primary vaccination series. Patients with chronic diseases were offered a booster vaccine starting from August 2021, with the recommendation to receive mRNA vaccines (Pfizer-BioNTech or Moderna) following non-mRNA primary vaccinations [23]. Individuals receiving only one dose of COVID-19 vaccine including the single-dose Janssen vaccine were excluded from our analysis due to inadequate primary immunisation.

### 2.5. Statistical Analysis

Continuous-type data were characterized by descriptive statistics such as number of samples (*n*) and mean ± standard deviation. Normality of the examined variables was checked with the Shapiro–Wilk test, and homogeneity of variances between the groups was checked with the Levene test. Data were found to be non-normally distributed and heteroscedastic. Categorical variables were presented as frequency and percentage or median with interquartile range (IQR). Pearson’s chi-square test and two-tailed Fisher’s exact test were applied for comparing categorical variables of GROUP A with GROUP B; and GROUP B with GROUP C. Continuous variables between subgroups were compared with the Kruskal–Wallis ANOVA with Bonferroni correction to determine significance between pairs of groups. In addition, survival probabilities of the three groups stratified by primary vaccination with different COVID-19 vaccines were displayed on a Kaplan–Meier plot and compared with a log-rank test. The analysis was two-sided with a significance level of α = 0.05. Statistical analysis was performed using the IBM^®^ SPSS^®^ 28.0 (IBM Corporation Armonk, NY, USA) program package.

## 3. Results

### 3.1. Baseline Characteristics

A total of 505 patients were distributed into three groups in this study. Patient characteristics are summarized in Table 2. Regarding gender, diabetes, hypertension, renal disease and cardiac disease, there were no significant differences between the three groups. However, the prevalence of cancer as a comorbidity was significantly higher in patients with cirrhosis compared to those without cirrhosis, with hepatocellular carcinoma (HCC) accounting for most cancer cases. Half of the 451 COVID-19 infected patients (227) were current smokers, and the proportion of smokers was found to be significantly different between the three groups. Ninety percent of all patients (453/505) received primary vaccination. As illustrated in Figure 2, mRNA-based COVID-19 vaccines were the most frequently administered in all three groups, particularly in GROUP B and GROUP C, but the rate of viral vector vaccination was higher in GROUP B than in GROUP C (11.5% vs. 7.4%).

### 3.2. Hospital Outcomes

The in-hospital mortality rates of the 505 patients divided into three groups were as follows: 11.8% (GROUP A); 9.6% (GROUP B); 11.8% (GROUP C). As demonstrated in Figure 3, the proportion of patients receiving oxygen support was lower in GROUP B compared to GROUP A. Nevertheless, 46% of patients in GROUP B had severe COVID-19 pneumonia with respiratory failure requiring mechanical ventilation during hospitalization. With regard to the pharmacological treatment of COVID-19, the administration of 5-day remdesivir was higher in GROUP A compared to GROUP B. Corticosteroids were the most commonly prescribed medications in both GROUP A and GROUP B, with rates of 71.9% and 71.2%, respectively. Convalescent plasma therapy was used in a slightly higher proportion in GROUP B (Figure 3).

### 3.3. Clinical Characteristics of Patients with Cirrhosis

The stage of cirrhosis was not significantly different between GROUP B and GROUP C (Table 2). In both groups, decompensated cirrhosis was the most frequent, with rates of 80.8% (GROUP B) and 59.3% (GROUP C), respectively. The prevalence of life-threatening ACLF in GROUP C was higher (14.8%) in comparison with GROUP B (11.5%), which may have contributed to higher in-hospital mortality in GROUP C (Figure 4). Regarding etiological factors of liver cirrhosis, alcohol was the leading cause, followed by HCV (Table 2). Moreover, the occurrence of autoimmune liver diseases including PBC, PSC and AIH was higher in GROUP B (13/52) compared to GROUP C (8/54).

Among the signs of hepatic decompensation, stage 3–4 encephalopathy was significantly more frequent in GROUP B (27/52) compared to GROUP C (18/54). As illustrated in Figure 4, grade 2–3 esophageal varices on endoscopy with higher risk of bleeding, indicating vascular decompensation, occurred more frequently in GROUP B (38/52) relative to GROUP C (29/54). As demonstrated in Figure 5, there was a stepwise increase in total cases with developing severity of cirrhosis in GROUP B classified using the Child–Turcotte–Pugh Score. In contrast, regarding developing cirrhosis severity, there was found to be a stepwise increase in mortality in GROUP C compared to GROUP B. However, proportions of total cases and deceased cases classified as CTP-A, CTP-B and CTP-C were not significantly different between GROUP B and GROUP C (Figure 5).

### 3.4. COVID-19 Vaccination

Results of Kaplan–Meier analysis of survival are depicted in Figure 6. As demonstrated in Figure 6A–C, patients who received primary immunization with mRNA vaccines had better survival outcomes compared to those vaccinated with viral vector or inactivated vaccines. Patients in GROUP A vaccinated with Moderna had a significantly better survival rate (log-rank test: *p* = 0.039) compared to those receiving Sputnik V (Figure 6A). Among mRNA vaccines, Pfizer-BioNTech was found to be significantly more efficient (log-rank test: *p* = 0.017) compared to Moderna (Figure 6C). Moreover, unvaccinated patients had worse hospital outcomes in all groups relative to those receiving any vaccine (Figure 6A–C). In addition, patients in GROUP C who did not receive a COVID-19 vaccine were significantly prone to increased in-hospital mortality (log-rank test: *p* = 0.003).

As demonstrated in Table 3, there were significant differences in major hospital outcomes including encephalopathy stage, ascites grade, esophageal varices on endoscopy, oxygen support and mechanical ventilation between the three groups stratified by mRNA vaccines.

Although the COVID-19 vaccination rate was equivalent in GROUP B and in GROUP C, both vaccinated and unvaccinated patients in GROUP B had a higher prevalence of stage 3–4 encephalopathy, moderate to severe ascites and grade 2–3 esophageal varices, due to hepatic decompensation and worsening liver cirrhosis. In addition, patients in GROUP B were significantly associated with O_2_ support and mechanical ventilation due to the onset of COVID-19-induced respiratory failure. More people were vaccinated with Pfizer-BioNTech in GROUP C compared to GROUP B (51.9% vs. 66.7%). Vaccination rates of Moderna in GROUP B and GROUP C were as follows: 23% (GROUP B) and 9.3% (GROUP C). As demonstrated in Table 3, patients vaccinated with Pfizer-BioNTech in GROUP C had significantly lower rates of worsening encephalopathy leading to fatal outcome in contrast to those in GROUP B (*p* < 0.05). Furthermore, primary vaccination with Moderna was significantly associated with in-hospital mortality in GROUP C compared to GROUP B (3.7% vs. 0%; *p* = 0.024). 

## 4. Discussion

Patients with liver cirrhosis are known to have a higher risk of in-hospital mortality and deterioration in cirrhosis severity following COVID-19 infection. Recently, multiple types of COVID-19 vaccines have been administered to the entire population including patients with chronic liver disease to prevent severe COVID-19 outcomes. We conducted a multicentre matched cohort study to investigate the efficacy of vaccines against SARS-CoV-2 in patients with liver cirrhosis following COVID-19 infection (GROUP B) compared to patients with liver cirrhosis without COVID-19 infection (GROUP C) and COVID-19 patients without cirrhosis (GROUP A).

In line with international data, our study demonstrated that patients in GROUP B were more frequently associated with adverse outcomes including oxygen support or mechanical ventilation compared to those in GROUP A (Figure 3) [24,25]. Previous studies showed that remdesivir was associated with liver injury in COVID-19 patients [26,27,28]. Moreover, Gao and colleagues reported that COVID-19 patients receiving corticosteroids had an increased risk of drug-induced liver injury (DILI) in contrast to non-user COVID-19 patients [29]. Therefore, in our dataset, patients receiving remdesivir or steroids were mostly patients in GROUP A (Figure 3). Consistently, patients in GROUP B received convalescent COVID-19 plasma (CCP) more commonly compared to those in GROUP A due to contraindications to start remdesivir in patients with five times the upper limit of transaminases. Higher administration rates of CCP in GROUP B might have resulted in improved outcomes, especially when the protein supplementation effect of this treatment is also considered (Figure 3) [30].

A study of a COVID-19 cohort of 220,727 US patients reported that hospital mortality rates of COVID-19 patients with cirrhosis and those without cirrhosis were 8.9% and 3.9%, respectively [25]. Regarding cirrhosis cases, the death rates in our study were not significantly different in the two groups: 9.6% (GROUP B) and 11.8% (GROUP C), respectively. Patients with cirrhosis had similarly poor prognosis regardless of COVID-19 infection, which was in accordance with a North American multicentre matched cohort study [7].

As for the etiology of cirrhosis, alcohol-related liver cirrhosis was found to be a leading cause and patients with alcohol-use disorder are notably more susceptible to hepatic decompensation following COVID-19 infection, in line with recently published studies [6,31,32].

The Clinical Practice Guidelines of the European Association for the Study of the Liver (EASL) emphasized the significance of regular clinical screening to the prompt detection and treatment of the complications of cirrhosis [33]. Surveillance procedures were likely delayed during the pandemic, leading to disease progression and increased occurrence of decompensation events. In our findings, patients with liver cirrhosis who contracted COVID-19 infection more frequently developed hepatic decompensation events with worsening ascites, significantly more severe encephalopathy stages and higher rates of acute variceal haemorrhage. Although viral infections could trigger ACLF, we found lower rates in GROUP B compared to GROUP C considering ACLF is mostly associated with bacterial infections and poor hospital outcome (Figure 4) [34,35]. According to a large registry cohort of 745 COVID-19 patients with chronic liver disease, liver cirrhosis severity classified by Child–Pugh score is a reliable predictor of in-hospital mortality [2]. We also found that disease progression in liver cirrhosis was precipitated by COVID-19 infection (Figure 5). Although there was a stepwise increase in cirrhosis progression following COVID-19 infection, GROUP B did not seem to be associated with higher mortality rates grouped by different CTP cirrhosis stages, in contrast to GROUP C. In accordance with international data, our current results demonstrate that patients with cirrhosis, without regard to COVID-19 status, remained at higher risk of in-hospital mortality [36]. 

A North American matched cohort of 762 patients reported that COVID-19-related in-hospital mortality was lower in patients with cirrhosis after receiving one or two mRNA vaccines in comparison with unvaccinated patients with liver cirrhosis [10]. Moreover, the two-dose administration of mRNA vaccines was associated with the highest efficacy of preventing a severe clinical course of COVID-19 infection compared to other vaccines [37]. In contrast to international data, our study showed that vaccines failed to prevent COVID-19-related hospitalisations in adults [9,38]. Nevertheless, in our survival analysis, unvaccinated patients, in particular those with liver cirrhosis, had significantly worse chances of survival (Figure 6). Regardless of COVID-19 status and liver cirrhosis, primary vaccination with mRNA vaccines, compared to viral vector or inactivated vaccines, significantly improved the survival rates, in accordance with previously published studies [39,40]. Comparing the effectiveness of primary vaccination with the two mRNA vaccines, Pfizer-BioNTech was found to be more effective in preventing symptomatic COVID-19 infection requiring hospital admission, and significantly decreased the need for oxygen support and mechanical ventilation in patients with cirrhosis (Table 3). Furthermore, primary vaccination with Pfizer-BioNTech in patients with cirrhosis significantly protected against decompensating events associated with hepatic encephalopathy. However, primary vaccination with Moderna in patients with cirrhosis was significantly associated with a poor outcome following COVID-19 infection compared to those vaccinated with Pfizer-BioNTech (Table 3).

Apart from mRNA vaccines, the Hungarian vaccination campaign included the AZD1222-AstraZeneca, Gam-COVID-Vac-Sputnik V and HB02-Sinopharm vaccines. In a nationwide, retrospective study investigating 3 740 066 Hungarian citizens, the overall estimated effectiveness of AstraZeneca, Sputnik-V and Sinopharm against COVID-19-related death was lower compared to Pfizer-BioNTech and Moderna [14]. Our study also found evidence that primary vaccination with viral vector and inactivated vaccines was less effective at preventing in-hospital mortality in COVID-19 patients, regardless of liver cirrhosis.

Altogether, our data provide evidence that COVID-19 infection is associated with the progression of liver cirrhosis and the development of acute hepatic decompensation events possibly due to aggravated immune dysfunction. Moreover, our findings demonstrate the effectiveness of primary vaccination with mRNA vaccines in patients with liver cirrhosis, predominantly in those receiving the Pfizer-BioNTech vaccine.

## 5. Limitations

The limitations of our data are mostly due to the retrospective study design and relatively small sample size. First, the number of enrolled patients with liver cirrhosis and those following COVID-19 infection were limited in our study. The higher mortality in GROUP C compared to GROUP B supports the notion that patients in GROUP C were admitted in a more vulnerable condition owing to advanced disease progression of cirrhosis. Second, the single-dose Ad26.COV2.S-Janssen vaccine was excluded from our investigation. As the cases were recorded over a prolonged time interval, a potential bias is expected as different cohorts were exposed to different contagion conditions. The study period overlapped with different waves of multiple variants of SARS-CoV-2, which may have influenced the effectiveness of the COVID-19 vaccines investigated in our analysis. Third, some vaccines were categorically indicated for elderly patients or patients with comorbidities, hence confounding patient characteristics could lead to perceived variation in the effectiveness of COVID-19 vaccines. Therefore, further prospective studies are needed to evaluate the effectiveness of all approved COVID-19 vaccines immunizing against new, upcoming variants in patients with cirrhosis.

On the other hand, the strengths of our study include its multicentre matched cohort design and its focus on a susceptible population group with a higher risk of COVID-19-related mortality. Although limited data are available about the impact of COVID-19 vaccination regimens on the clinical outcome of patients with liver cirrhosis, our methodology exclusively allows better interpretation to evaluate the impact of different COVID-19 vaccines on hospital outcomes in COVID-19 infection and liver cirrhosis.

## 6. Conclusions

Patients with cirrhosis experiencing a COVID-19 infection requiring hospitalization were significantly coupled with acute hepatic decompensation events. However, baseline liver cirrhosis severity is a major determinant of in-hospital mortality. Hence, primary vaccination with mRNA vaccines was significantly associated with better a prognosis in patients with cirrhosis. Outstandingly, primary vaccination with the BNT162b2 vaccine was the most effective in preventing acute hepatic decompensating events, COVID-19-related adverse outcomes and in-hospital mortality.

## Figures and Tables

**Figure 1 vaccines-11-00050-f001:**
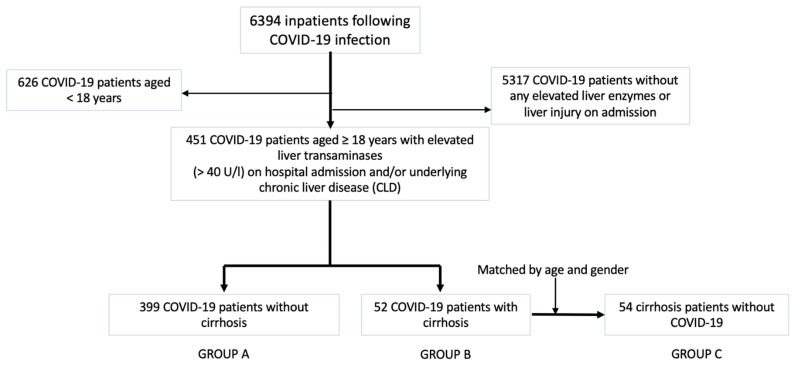
The flowchart of patient enrolment and cohort selection.

**Figure 2 vaccines-11-00050-f002:**
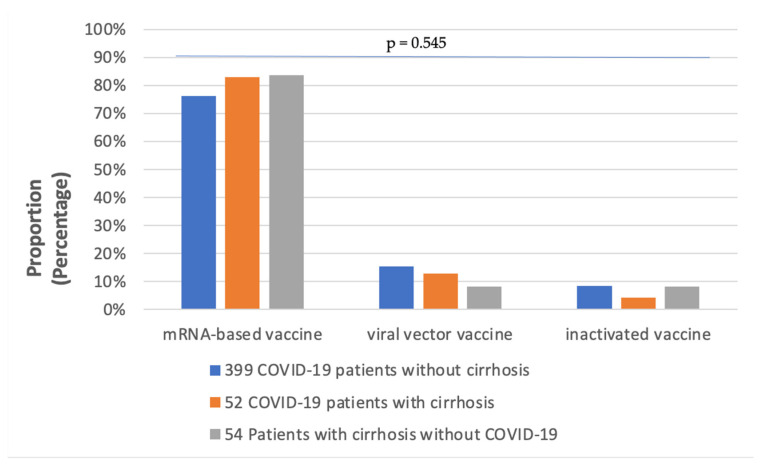
The rates of patients receiving different COVID-19 vaccines in the 3 groups.

**Figure 3 vaccines-11-00050-f003:**
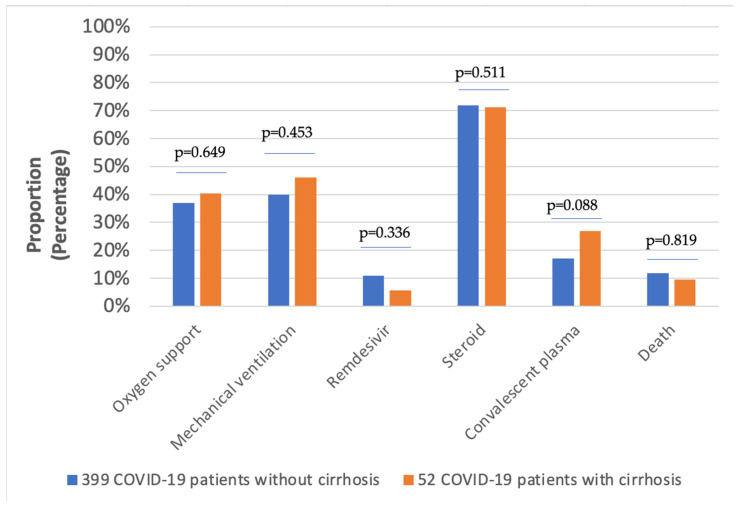
Treatment and hospital outcomes in COVID-19 patients.

**Figure 4 vaccines-11-00050-f004:**
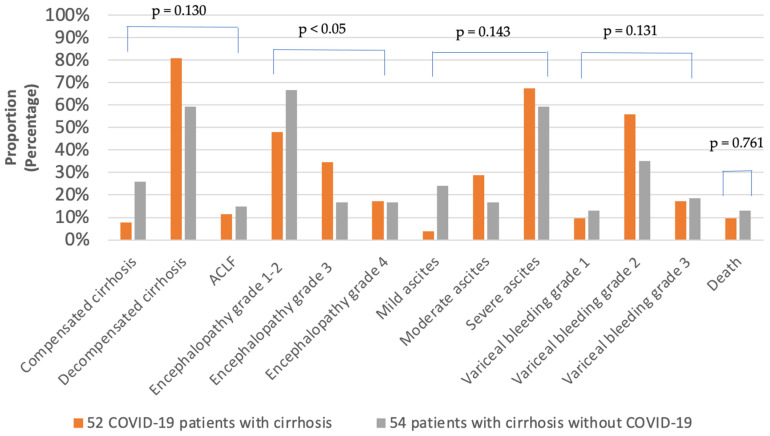
Clinical characteristics of patients with liver cirrhosis.

**Figure 5 vaccines-11-00050-f005:**
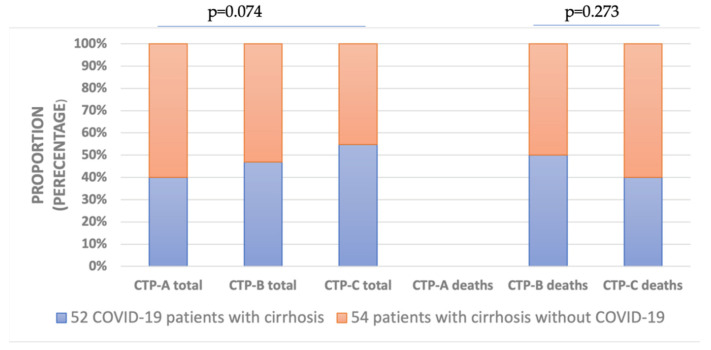
Proportion of different Child–Turcette–Pugh (CTP) stages in all and deceased patients with cirrhosis. There were no CTP-A deceased cases in both groups. The severity of liver cirrhosis was classified by CTP Score.

**Figure 6 vaccines-11-00050-f006:**
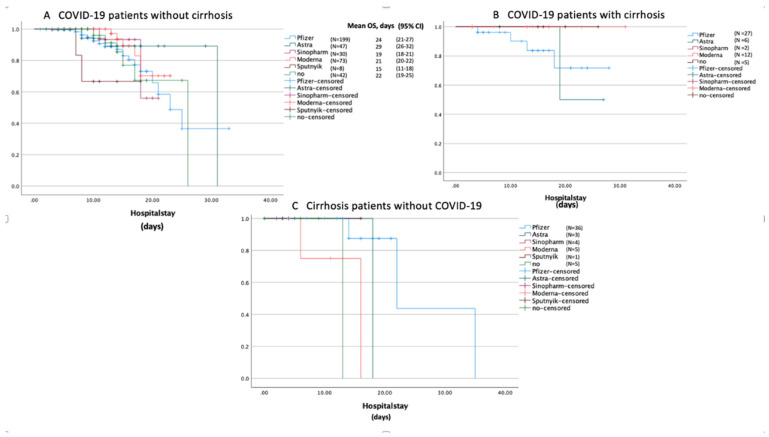
Kaplan–Meier survival curves displaying the estimated survival probabilities in the three groups (**A**–**C**) stratified by different COVID-19 vaccines. Mean overall survival (OS) and 95% confidence intervals (95% CI) are included, where appropriate. (**A**) In COVID-19 patients without cirrhosis, Sputnik-V was associated with significantly lower survival rates compared to Moderna (log-rank test: *p* = 0.039). (**B**) In COVID-19 patients with cirrhosis, AstraZeneca was less effective to prevent COVID-19-related deaths in contrast to Moderna (log-rank test: *p* = 0.157). (**C**) Patients receiving Pfizer-BioNTech had significantly better survival rates compared to those receiving Moderna (log-rank test: *p* = 0.017).

**Table 1 vaccines-11-00050-t001:** Inclusion and exclusion criteria.

Patient Group	Inclusion Criteria	Exclusion Criteria
GROUP A and GROUP B	Laboratory-confirmed SARS-CoV-2 infection Age ≥ 18 years Elevated liver transaminases (>40 U/L) on hospital admission and/or underlying chronic liver disease (CLD) in the medical history	Rapid Antigen Test-confirmed SARS-CoV-2 infection without PCR-positivity Age < 18 years
GROUP A	Absence of liver cirrhosis diagnosis	Patients receiving one dose of COVID-19 vaccine including single-dose Janssen vaccine
GROUP B	Clinicopathologically confirmed liver cirrhosis	
GROUP C	Clinicopathologically confirmed liver cirrhosis; at least 2 days of hospitalisation in hepatology units; matched with GROUP B for equivalent severity grades and clinical course	
	Laboratory-confirmed SARS-CoV-2 negativity	SARS-CoV-2 positivity on admission
	Age ≥ 18 years	Age < 18 years

**Table 2 vaccines-11-00050-t002:** Comparison of baseline characteristics across groups.

Variables	GROUP A (n = 399)	GROUP B (n = 52)	GROUP C (n = 54)	*p**	*p* ^-^	*p* ^;^
Gender (male/female)	219/180	36/16	33/21	0.121	0.054	0.420
Fatal outcome	47 (11.8)	5 (9.6)	7 (13.0)	0.880	0.819	0.761
Liver disease	46 (11.5)	52 (100)	54 (100)	**<0.001**	**<0.001**	-
Stage of cirrhosis Compensated Decompensated ACLF	-	4 (7.7) 42 (80.8) 6 (11.5)	14 (25.9) 32 (59.3) 8 (14.8)	**<0.001**	**<0.001**	0.130
Cirrhosis etiology Alcohol PBC PSC AIH HBV HCV Cryptogen NASH Budd-Chiari Wilson’s disease Haemochromatosis Cystic fibrosis	3 (0.8) 1 (0.3) 5 (1.3) 0 (0) 1 (0.3) 2 (0.6) 0 (0) 0 (0) 0 (0) 1 (0.3) 1 (0.3) 0 (0)	27 (51.9) 2 (3.8) 6 (11.5) 5 (9.6) 2 (3.8) 8 (15.4) 3 (5.8) 0 (0) 1 (1.9) 0 (0) 0 (0) 0 (0)	22 (40.7) 0 (0) 5 (9.3) 3 (5.6) 4 (7.4) 10 (18.6) 8 (14.8) 1 (1.9) 0 (0) 1 (1.9) 1 (1.9) 1 (1.9)	**<0.001** **<0.05** **<0.001** **<0.001** **<0.001** **<0.001** **<0.001** **<0.05** 0.103 **<0.05** **<0.05** **<0.05**	**<0.001** **<0.05** **<0.001** **<0.001** **<0.05** **<0.001** **<0.05** 0.692 0.115 0.885 0.885 -	0.330 0.238 0.759 0.484 0.679 0.797 0.202 0.509 0.491 0.509 0.509 0.509
Ascites grades Mild Moderate Severe	0 (0) 0 (0) 0 (0)	2 (3.8) 15 (28.8) 35 (67.4)	13 (24) 9 (16.7) 32 (59.3)	**<0.001**	**<0.001**	0.143
Encephalopathy stages 1-2 3 4	398 (99.7) 1 (0.3) 0 (0)	25 (48.0) 18 (34.5) 9 (17.3)	36 (66.7) 9 (16.7) 9 (16.7)	**<0.001**	**<0.001**	**<0.05**
Erosive oesophagitis	93 (23.3)	31 (59.6)	41 (75.9)	**<0.001**	**<0.001**	0.096
Esophageal varices Grade 1 Grade 2 Grade 3	2 (0.6) 1 (0.3) 0 (0)	5 (9.6) 29 (55.8) 9 (17.3)	7 (13.0) 19 (35.2) 10 (18.5)	**<0.001**	**<0.001**	0.131
Hypertension	239 (59.9)	29 (55.8)	32 (59.3)	0.882	0.653	0.844
Cardiovascular disease	172 (43.1)	22 (42.3)	29 (53.7)	0.337	0.518	0.251
Diabetes mellitus	133 (33.3)	20 (38.5)	20 (37.0)	0.700	0.534	0.519
Renal disease	60 (15.0)	6 (11.5)	8 (14.8)	0.810	0.544	0.776
Cancer HCC CRC Pancreas Klatskin	29 (7.3) 18 (4.5) 5 (1.3) 3 (0.8) 1 (0.3)	9 (17.3) 6 (11.5) 2 (3.8) 0 (0) 1 (1.9)	9 (16.7) 9 (16.7) 0 (0) 0 (0) 0 (0)	**<0.05**	**<0.05**	0.795
Smoking	202 (50.6)	25 (48.0)	17 (31.5)	**<0.05**	0.769	0.112
COVID-19 Treatment Remdesivir Steroid use Convalescent Plasma	44 (11) 287 (71.9) 68 (17.0)	3 (5.8) 37 (71.2) 14 (26.9)	0 (0) 0 (0) 0 (0)	**<0.05** **<0.001** **<0.001**	0.336 0.511 0.088	0.115 **<0.001** **<0.001**
Oxygen supply	147 (41)	21 (40.4)	3 (5.6)	**<0.001**	0.649	**<0.001**
Mechanical ventilation	159 (39.8)	24 (46.2)	6 (11.1)	**<0.001**	0.453	**<0.001**
COVID-19 vaccination	357 (89.5)	47 (90.4)	49 (90.7)	0.966	0.535	0.605
COVID-19 vaccines mRNA viral vector inactivated	272 (68.2) 55 (13.8) 30 (7.5)	39 (75.0) 6 (11.5) 2 (3.8)	41 (75.9) 4 (7.4) 4 (7.4)	0.545	0.818	0.171

Categorical variables are presented as frequency (percentage). Bold text highlights the statistically significant values. GROUP A: 399 COVID-19 patients without cirrhosis. GROUP B: 52 COVID-19 patients with cirrhosis. GROUP C: 54 patients with cirrhosis without COVID-19 infection. *p**: Kruskal–Wallis, chi-square and Analysis of Variance (ANOVA) as appropriate. *p*^-^ < 0.05 statistically significant between GROUP A and GROUP B. *p*^;^ < 0.05 statistically significant between GROUP B and GROUP C. Abbreviations: ACLF, acute-on-chronic liver failure; PBC, Primary Biliary Cholangitis; PSC, Primary Sclerosing Cholangitis; AIH, Autoimmune Hepatitis; HBV, Hepatitis B Virus; HCV, Hepatitis C Virus; NASH, nonalcoholic steatohepatitis; HCC, hepatocellular cancer; CRC, colorectal cancer; COVID-19, Coronavirus disease 2019; mRNA, messenger RNA.

**Table 3 vaccines-11-00050-t003:** The impact of primary vaccination with mRNA vaccines on hospital outcomes regarding COVID-19 status and cirrhosis.

Variable	GROUP B (*n* = 52)		GROUP A (*n* = 399)		GROUP C (*n* = 54)		*p**		*p*-		*p*;	
	Pfizer-BioNTech (*n* = 27)	Moderna (*n* = 12)	Pfizer-BioNTech (*n* = 199)	Moderna (*n* = 73)	Pfizer-BioNTech (*n* = 36)	Moderna (*n* = 5)	Pfizer-BioNTech	Moderna	Pfizer-BioNTech	Moderna	Pfizer-BioNTech	Moderna
**Encephalopathy stage**							**<0.001**	**<0.001**	**<0.001**	**<0.001**	**<0.05**	0.082
**Stage 1-2**	11 (21.2	6 (11.5)	199 (49.9)	72 (18)	25 (46.3)	1 (1.9)						
**Stage 3**	8 (15.4)	6 (11.5)	0 (0)	1 (0.3)	7 (13)	2 (3.7)						
**Stage 4**	8 (15.4)	0 (0)	0 (0)	0 (0)	4 (7.4)	2 (3.7)						
**Ascites grade**							**<0.001**	**<0.001**	**<0.001**	**<0.001**	0.094	0.503
**Mild**	0 (0)	0 (0)	199 (49.9)	72 (18)	10 (18.5)	0 (0)						
**Moderate**	6 (11.5)	5 (9.6)	0 (0)	1 (0.3)	3 (5.6)	3 (5.6)						
**Severe**	21 (40.4)	7 (13.5)	0 (0)	0 (0)	23 (42.6)	2 (3.7)						
**Esophageal varices**							**<0.001**	**<0.001**	**<0.001**	**<0.001**	0.162	0.412
**Grade 1**	3 (5.8)	0 (0)	1 (0.3)	1 (0.3)	4 (7.4)	0 (0)						
**Grade 2**	15 (28.8)	8 (15.4)	1 (0.3)	0 (0)	14 (25.9)	1 (1.9)						
**Grade 3**	6 (11.5)	1 (1.9)	0 (0)	0 (0)	7 (13)	1 (1.9)						
**Oxygen support**	9 (17.3)	6 (11.5)	65 (16.3)	29 (7.3)	1 (1.9)	1 (1.9)	**<0.05**	0.517	0.945	0.505	**<0.05**	0.267
**Mechanical ventilation**	16 (30.8)	4 (7.7)	89 (22.3)	22 (5.5)	5 (9.3)	1 (1.9)	**<0.001**	0.861	0.156	0.825	**<0.001**	0.594
**Fatal outcome**	4 (7.7)	0 (0)	25 (6.3)	6 (1.5)	3 (5.6)	2 (3.7)	0.706	**<0.05**	0.743	0.306	0.422	**<0.05**

Categorical variables are presented as frequency (percentage). Bold text highlights the statistically significant values. GROUP A: 399 COVID-19 patients without cirrhosis. GROUP B: 52 COVID-19 patients with cirrhosis. GROUP C: 54 patients with cirrhosis without COVID-19 infection. *p**: Kruskal–Wallis, chi-square and analysis of variance as appropriate among the 3 groups. *p*^-^ < 0.05 statistically significant between GROUP A and GROUP B receiving different mRNA vaccines. *p***^;^** < 0.05 statistically significant between GROUP B and GROUP C receiving different mRNA vaccines.

## Data Availability

Accessible upon reasonable request from the corresponding author.

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
