# Peer review of "Effectiveness of COVID-19 Vaccination with mRNA Vaccines for Patients with Cirrhosis in Hungary: Multicentre Matched Cohort Study"

_vaccines, 2022, doi:10.3390/vaccines11010050_

Round 1

Reviewer 1 Report

This paper discusses statistics related to cirrhosis, COVID, and vaccination status in Hungary.

This paper was a bit of a tough read, both because the English was rough, and because the logic is not well laid out. Nevertheless one has the sense that the data might support some interesting conclusions, if such conclusions were presented more compellingly.

I think the paper is best understood by reading the CONCLUSIONS first:

"Patients with cirrhosis experiencing a COVID-19 infection needing hospitalization were significantly coupled with acute hepatic decompensation events. However, baseline liver cirrhosis severity is a major determinant of in-hospital mortality. Hence, primary vaccination with mRNA vaccines were significantly associated with better prognosis in patients with cirrhosis. Outstandingly primary vaccination with BNT162b2 vaccine was the most effective in preventing acute hepatic decompensating events, COVID-19 related adverse outcomes and in-hospital mortality."

In other words, the authors are recommending that patients with cirrhosis get vaccinated. And if they have a choice of vaccines they should get BNT162b2.

Consider going back through the ENTIRE paper with this as a guiding light and write the paper to clearly focus on reaching this conclusion. Consider moving other tangential analyses to supplemental so that you can tell a clear story.

For example, do we really care what vaccine people in GROUP A or in GROUP C got? Those groups are not important for making one of the main points that you are trying to make in the conclusions, which is focused solely on what vaccine is best for folks with cirrhosis.

Another option would be to drop most of the discussion of vaccination and focus on the difference between GROUP A and GROUP B. And try to focus on writing a paper about how to care for hospitalized folks with both cirrhosis and COVID.

MAJOR

In the Abstract state very clearly your hypothesis, and how you tested it. This is currently very confusing. Strongly consider using a structured Abstract, if the journal allows it. If not, write your abstract as a structured abstract, and then gently remove the structure, leaving the logic intact. See if you can find an appropriate EQUATOR guideline (take a look at these as most of these are relevant: https://www.equator-network.org/reporting-guidelines/consort/) to help you understand what elements are important to reporting on a matched cohort study.

What is the reason each group (A,B,C) was hospitalized? Was it because of COVID or cirrhosis? One assumes mortality rate is largely driven by the reason for hospitalization.

Did group A & B get COVID before or during their hospitalization?

When was each vaccine primarily available in Hungary? Were they are equally available at all months of all years? Or did the vaccines become available at different times? If they did, then survival and outcomes may be related to many other factors, such as SARS-CoV-2 variant and improving ability to treat COVID.

Are there other factors that might be associated with vaccine choice, such as race or socioeconomic class?

Is the vaccination important in keeping people out of the hospital? If so, you are asking whether vaccination is good or bad GIVEN THAT THE VACCINE ALREADY FAILED TO KEEP THIS PARTICULAR INDIVIDUAL OUT OF THE HOSPITAL. You should probably mention this, and discuss how it affects interpretation of your data.

MINOR

TITLE

Title too wordy

"Effectiveness of primary vaccination with mRNA vaccines in patients with cirrhosis without COVID-19 infection and COVID-19 patients with cirrhosis: multicentre matched cohort study"

->

"Effectiveness of COVID-19 vaccination with mRNA vaccines for patients with cirrhosis in Hungary: multicentre matched cohort study"

ABSTRACT

list group A before group B before group c. 

or maybe better: don't use the "GROUP" terminology in the abstract. It is confusing and hard to follow. Just write out what you mean in English.

______

"Of cirrhosis cases, we found higher mortality rates in GROUP C (11.8%) compared to GROUP B (9.6%)"

specify confidence intervals for each of these numbers, and whether the difference is significant. If it is not significant rewrite the sentence (e.g., if p = 0.07) as

"Of cirrhosis cases, we found no difference (p= 0.07) in mortality rates in GROUP C (11.8%) compared to GROUP B (9.6%)"

If p<0.05 then explain why you think COVID-19 makes people healthier.

______

"Overall, GROUP C was predominantly vaccinated with BNT162b2 vaccine resulting in significantly lower rates of oxygen support and mechanical ventilation."

Is there a difference in what vaccine each group was vaccinated with?

KEYWORDS

COVID-19 & COVID-19 vaccination are a bit redundant. Consider just COVID-19 & vaccination. Or just the single "COVID-19 vaccination"

INTRODUCTION

Introduce all acronyms (e.g., CLD) the first time they are used.

"Despite acute hepatic decompensation and ACLF have been associated with SARS-CoV-2, the leading cause of death is up to now the respiratory failure (71%) and subsequently the liver-related manifestations (19%) [2]."

This sentence is hard to understand due to poor English grammar. You may need to have an expert in English help you re-write your paper to make sure your message is communicated. Also consider writing several shorter sentences in places where you are trying to communicate more than one concept in a sentence. The above sentence could probably be three short sentences. Writing short sentences can help with writing clear meaning, particularly if you are not an English expert.

"Patients with chronic hepatitis virus (HBV, HCV) without cirrhosis were not found to be at increased exposure to in-hospital COVID-19 mortality [9,10]."

This sentence is awkward. One is not "exposed" to mortality. I am forced to guess at your meaning. I think I know what you mean, but am not sure. And I am a bit of an expert in the area, so considerable rewriting needs to be done to allow folks outside of your immediate area to understand this paper.

"To date, there are limited data about the efficacy of different vaccines against SARS-CoV-2 (mRNA; viral vector; inactivated) in patients with cirrhosis following COVID-19 infection compared to those without COVID-19."

This seems to be a key sentence in hinting at what your aim and hypothesis is for this paper. But I don't get it. Are you proposing to vaccinate cirrhotic patients AFTER they get COVID?

METHODS

"A total of 6394 COVID-19 patients were recruited in our analysis"

but then you state that you find "54 cirrhosis patients without COVID-19 infection"

where do those come from?

Figure 1 makes it seem that these 54 are a subset of the 52 in GROUP B, which is impossible, as 54>52.

FIGURE 2

consider a "stacked column chart" (you did this for Figure 5)

also consider NOT using "GROUP A, B, C" but rather spelling out what each group is. A lot of readers just read the figures and not the text, so these readers will have no idea what GROUP B is. Maybe you sort of did that in the legend, but it is very oddly formatted.

FIGURE 3

need to put in p-values for significances. Please add confidence and statistics throughout your manuscript, where appropriate.

FIGURE 6

the legends seem to have duplicated labels. I only see the trend lines for "-censored" not the other square-wave trend lines.

"As demonstrated in Figure 6 A, patients receiving primary vaccination of mRNA vaccines had better survival outcomes compared to those vaccinated with viral vector or inactivated vaccines."

Figure 6A DOES NOT 'demonstrate' this. First, you have not indicated which lines are viral vector and which are inactivated. Some colors are hard to tell apart (e.g., Sputnyk and Sinopharm). And you have not included any statistics. Most of the trajectories look pretty similar to me.

FIGURE 6

you have group B in panel B and Group A in panel C. A comes before B in the alphabet.

FIGURE 6

Add N if possible, if not in the figures, then at least in the legend.

Section Numbering

Discussion, Limitations, Conclusions are all "Section 1"

Author Response

Response to Reviewer 1 Comments

Point 1: This paper was a bit of a tough read, both because the English was rough, and because the logic is not well laid out. Nevertheless one has the sense that the data might support some interesting conclusions, if such conclusions were presented more compellingly.

I think the paper is best understood by reading the CONCLUSIONS first:

"Patients with cirrhosis experiencing a COVID-19 infection needing hospitalization were significantly coupled with acute hepatic decompensation events. However, baseline liver cirrhosis severity is a major determinant of in-hospital mortality. Hence, primary vaccination with mRNA vaccines were significantly associated with better prognosis in patients with cirrhosis. Outstandingly primary vaccination with BNT162b2 vaccine was the most effective in preventing acute hepatic decompensating events, COVID-19 related adverse outcomes and in-hospital mortality."

In other words, the authors are recommending that patients with cirrhosis get vaccinated. And if they have a choice of vaccines they should get BNT162b2. 

Consider going back through the ENTIRE paper with this as a guiding light and write the paper to clearly focus on reaching this conclusion. Consider moving other tangential analyses to supplemental so that you can tell a clear story.

For example, do we really care what vaccine people in GROUP A or in GROUP C got? Those groups are not important for making one of the main points that you are trying to make in the conclusions, which is focused solely on what vaccine is best for folks with cirrhosis.

Another option would be to drop most of the discussion of vaccination and focus on the difference between GROUP A and GROUP B. And try to focus on writing a paper about how to care for hospitalized folks with both cirrhosis and COVID.

Response 1: The Reviewer has an important point. We rewrote the entire paper including abstract, introduction, material and methods, results, discussion and conclusions to reach this above-mentioned conclusion focusing on which vaccine is best for folks with cirrhosis.

MAJOR

Point 2: In the Abstract state very clearly your hypothesis, and how you tested it. This is currently very confusing. Strongly consider using a structured Abstract, if the journal allows it. If not, write your abstract as a structured abstract, and then gently remove the structure, leaving the logic intact. See if you can find an appropriate EQUATOR guideline (take a look at these as most of these are relevant: https://www.equator-network.org/reporting-guidelines/consort/) to help you understand what elements are important to reporting on a matched cohort study.

 Response 2: We fully agree with the Reviewer. We rewrote it as a structured abstract according to EQUATOR guideline as follows:

„Patients with cirrhosis are vulnerable to hepatic decompensation events and death following COVID-19 infection. Therefore, primary vaccination with COVID-19 vaccines is fundamental to reducing risk of COVID-19 related deaths in patients with cirrhosis. However, limited data is available about the effectiveness of mRNA vaccines compared to other vaccines. The aim of our study was to investigate the efficacy of mRNA vaccines versus other vaccines in cirrhosis. In this retrospective study, we compared clinical characteristics and vaccine effectiveness of 399 COVID-19 patients without cirrhosis (GROUP A) to 52 COVID-19 patients with cirrhosis (GROUP B). 54 hospitalised cirrhosis controls without COVID-19 (GROUP C) were randomly sampled 1:1 and matched by gender and age. Of cirrhosis cases, we found no difference (p=0.761) in mortality rates in controls without COVID-19 (11.8%) compared to those with COVID-19 (9.6%). However, COVID-19 patients with cirrhosis were associated with higher rates of worsening hepatic encephalopathy, ascites and esophageal varices. Patients with cirrhosis receiving mRNA vaccines had significantly better survival rates compared to viral vector or inactivated vaccines. Outstandingly primary vaccination with BNT162b2 vaccine was the most effective in preventing acute hepatic decompensating events, COVID-19 infection requiring hospital admission and in-hospital mortality.”

Point 3: What is the reason each group (A,B,C) was hospitalized? Was it because of COVID or cirrhosis? One assumes mortality rate is largely driven by the reason for hospitalization.

Response 3: The Reviewer has a good point. As demonstrated in Table 1, patient groups (A,B) were hospitalised due to laboratory-confirmed COVID-19 infection. GROUP C cohort was admitted due to deterioration in cirrhosis stage (Line 81-85).

“The cirrhosis controls without COVID-19 cohort included all laboratory-confirmed SARS-CoV-2 negative, adult patients who were previously diagnosed with liver cirrhosis and were admitted in the participating centres due to deterioration in cirrhosis stage between March 2020 and May 2022.”  

Point 4: Did group A & B get COVID before or during their hospitalization?

Response 4: The Reviewer has an important point. GROUP A and GROUP B were laboratory-confirmed COVID-19 cases on hospital admission.

 „We included 451 COVID-19 patients aged ≥ 18 years with elevated liver transaminases (> 40 U/l) on hospital admission and/or underlying chronic liver disease (CLD) in the medical history.”

Point 5: When was each vaccine primarily available in Hungary? Were they are equally available at all months of all years? Or did the vaccines become available at different times? If they did, then survival and outcomes may be related to many other factors, such as SARS-CoV-2 variant and improving ability to treat COVID.

Response 5: We fully agree with the Reviewer. We incluced availability of vaccines in Hungary as follows:

“In Hungary, Pfizer-BioNTech was the first vaccine to be used in cirrhosis. Primary vaccination campaign using mRNA vaccines (Pfizer-BioNTech and Moderna) started on 15 January 2021. During the third wave in March and April 2021, five different vaccines were widely used in cirrhosis: two mRNA-based vaccines (BNT162b2-Pfizer-BioNTech and mRNA-1273-Moderna, two vector vaccines (AZD1222-Astra Zeneca, Gam-COVID-Vac-Sputnik V) and one inactivated vaccine (HB02-Sinopharm) [14].”

Point 6: Are there other factors that might be associated with vaccine choice, such as race or socioeconomic class?

Response 6: The Reviewer has a good point. According to international data, socioeconomic-status such as high deprivation is associated with poor prognosis in COVID-19 infection. Therefore, patients in high deprivation need to be vaccinated with more efficient mRNA vaccines.

Point 7: Is the vaccination important in keeping people out of the hospital? If so, you are asking whether vaccination is good or bad GIVEN THAT THE VACCINE ALREADY FAILED TO KEEP THIS PARTICULAR INDIVIDUAL OUT OF THE HOSPITAL. You should probably mention this, and discuss how it affects interpretation of your data.

Response 7: The Reviewer has an important point. We included this point in our study as follows (Line 333-334):

“In contrast to international data, our study showed that vaccines failed to prevent COVID-19 related hospitalisations in adults [9,38].”

MINOR

TITLE

Point 8: Title too wordy

"Effectiveness of primary vaccination with mRNA vaccines in patients with cirrhosis without COVID-19 infection and COVID-19 patients with cirrhosis: multicentre matched cohort study"

"Effectiveness of COVID-19 vaccination with mRNA vaccines for patients with cirrhosis in Hungary: multicentre matched cohort study"

 Response 8: We fully agree with the Reviewer. We reformulated the title as follows.
"Effectiveness of COVID-19 vaccination with mRNA vaccines for patients with cirrhosis in Hungary: multicentre matched cohort study" 

Point 9: ABSTRACT

list group A before group B before group c. 

or maybe better: don't use the "GROUP" terminology in the abstract. It is confusing and hard to follow. Just write out what you mean in English.

"Of cirrhosis cases, we found higher mortality rates in GROUP C (11.8%) compared to GROUP B (9.6%)"

specify confidence intervals for each of these numbers, and whether the difference is significant. If it is not significant rewrite the sentence (e.g., if p = 0.07) as

"Of cirrhosis cases, we found no difference (p= 0.07) in mortality rates in GROUP C (11.8%) compared to GROUP B (9.6%)"

If p<0.05 then explain why you think COVID-19 makes people healthier.

"Overall, GROUP C was predominantly vaccinated with BNT162b2 vaccine resulting in significantly lower rates of oxygen support and mechanical ventilation."

Is there a difference in what vaccine each group was vaccinated with?

Response 9:

The Reviewer has good points. We reformulated the order of groups in the abstract as follows:

“In this retrospective study, we compared clinical characteristics and vaccine effectiveness of 399 COVID-19 patients without cirrhosis (GROUP A) to 52 COVID-19 patients with cirrhosis (GROUP B). 54 hospitalised cirrhosis controls without COVID-19 (GROUP C) were randomly sampled 1:1 and matched by gender and age.”

The difference in mortality rates between GROUP C compared to GROUP B was not significant. Therefore, we reformulated the sentence as follows:

“Of cirrhosis cases, we found no difference (p=0.761) in mortality rates in controls without COVID-19 (11.8%) compared to those with COVID-19 (9.6%).”

In the abstract, we focused on GROUP B instead of GROUP C, hence we excluded the following sentence: "Overall, GROUP C was predominantly vaccinated with BNT162b2 vaccine resulting in significantly lower rates of oxygen support and mechanical ventilation."

Point 10: KEYWORDS

COVID-19 & COVID-19 vaccination are a bit redundant. Consider just COVID-19 & vaccination. Or just the single "COVID-19 vaccination"

Response 10: The Reviewer has a good point: Keywords were reformulated as follows:

“liver cirrhosis; COVID-19 vaccination; mRNA vaccines; vaccine effectiveness”

Point 11: INTRODUCTION

Introduce all acronyms (e.g., CLD) the first time they are used.

"Despite acute hepatic decompensation and ACLF have been associated with SARS-CoV-2, the leading cause of death is up to now the respiratory failure (71%) and subsequently the liver-related manifestations (19%) [2]."

This sentence is hard to understand due to poor English grammar. You may need to have an expert in English help you re-write your paper to make sure your message is communicated. Also consider writing several shorter sentences in places where you are trying to communicate more than one concept in a sentence. The above sentence could probably be three short sentences. Writing short sentences can help with writing clear meaning, particularly if you are not an English expert.

"Patients with chronic hepatitis virus (HBV, HCV) without cirrhosis were not found to be at increased exposure to in-hospital COVID-19 mortality [9,10]."

This sentence is awkward. One is not "exposed" to mortality. I am forced to guess at your meaning. I think I know what you mean, but am not sure. And I am a bit of an expert in the area, so considerable rewriting needs to be done to allow folks outside of your immediate area to understand this paper.

"To date, there are limited data about the efficacy of different vaccines against SARS-CoV-2 (mRNA; viral vector; inactivated) in patients with cirrhosis following COVID-19 infection compared to those without COVID-19."

This seems to be a key sentence in hinting at what your aim and hypothesis is for this paper. But I don't get it. Are you proposing to vaccinate cirrhotic patients AFTER they get COVID?

 Response 11: The Reviewer has important point. We introduced the acronym (e.g., CLD) the first time it was used as follows:

“Furthermore, patients with decompensated cirrhosis following SARS-CoV-2 infection were more susceptible to Intensive Care Unit (ICU) admission, renal replacement therapy and invasive mechanical ventilation compared to those without chronic liver disease (CLD) [6]. Nevertheless lung injury and hepatic dysfunctions are tied by multiple links such as an altered gut-lung axis, gut dysbiosis, cirrhosis-associated immune dysfunction (CAID) and pulmonary complications due to ascites [8]”.

 Our paper is rewritten by an English expert to make sure our message is communicated. We also considered writing several shorter sentences as follows:

“Despite being respiratory failure the leading cause (71%) in COVID-19, liver-related manifestations (19%) are also common [2].”

We excluded that awkward sentence from our study because vaccine effectiveness is the main point in our analysis.

Regarding patients with cirrhosis getting vaccinations, cirrhosis patients received primary immunization before COVID-19 infection.

Point 12: METHODS

"A total of 6394 COVID-19 patients were recruited in our analysis"

but then you state that you find "54 cirrhosis patients without COVID-19 infection"

where do those come from?

Response 12: The Reviewer has a good point. We reformulated Section 2.1 Patient cohorts as follows:

“Thus, GROUP B patients were matched with 54 cirrhosis patients without COVID-19 infection (GROUP C) in respect to age and gender in 1:1 ratio (Figure 1). The cirrhosis controls without COVID-19 cohort included all laboratory-confirmed SARS-CoV-2 negative, adult patients who were previously diagnosed with liver cirrhosis and were admitted in the participating centres due to deterioration in cirrhosis stage between March 2020 and May 2022.”

Point 13: Figure 1 makes it seem that these 54 are a subset of the 52 in GROUP B, which is impossible, as 54>52.

Response 13: We fully agree with the Reviewer. We included how the matching was performed and on which characteristica.

Point 14: FIGURE 2

consider a "stacked column chart" (you did this for Figure 5)

also consider NOT using "GROUP A, B, C" but rather spelling out what each group is. A lot of readers just read the figures and not the text, so these readers will have no idea what GROUP B is. Maybe you sort of did that in the legend, but it is very oddly formatted.

Response 14: The Reviewer has a good point. We included groups spelling out instead of using GROUP A, B, C.

Point 15: FIGURE 3

need to put in p-values for significances. Please add confidence and statistics throughout your manuscript, where appropriate.

 Response 15: We fully agree with the Reviewer. We put in p-values for significances in Figure 2-5 and mean overall estimates with confidence intervals in Figure 6, where appropriate.

Point 16: FIGURE 6

the legends seem to have duplicated labels. I only see the trend lines for "-censored" not the other square-wave trend lines. 

"As demonstrated in Figure 6 A, patients receiving primary vaccination of mRNA vaccines had better survival outcomes compared to those vaccinated with viral vector or inactivated vaccines."

Figure 6A DOES NOT 'demonstrate' this. First, you have not indicated which lines are viral vector and which are inactivated. Some colors are hard to tell apart (e.g., Sputnyk and Sinopharm). And you have not included any statistics. Most of the trajectories look pretty similar to me.

 Response 16: We fully agree with the Reviewer. We excluded Figure 6A from our study. Moreover, we provided mean overall survival and 95% confidence intervals in Figure 2A.

Point 17: FIGURE 6

you have group B in panel B and Group A in panel C. A comes before B in the alphabet.

Response 17: The Reviewer has a good point. We made changes in the legend as follows:

„A) In COVID-19 patients without cirrhosis, Sputnik-V was associated with significantly lower survival rates compared to Moderna (log rank test: p=0.039). (B) In COVID-19 patients with cirrhosis, AstraZeneca was less effective to prevent COVID-19 related deaths in contrast to Moderna (log-rank test: p=0.157). (C) Patients receiving Pfizer-BioNTech had significantly better survival rates compared to those receiving Moderna (log-rank test: p=0.017).”  

Point 18: FIGURE 6

Add N if possible, if not in the figures, then at least in the legend.

Response 18: The Reviewer has a good point. We added N (numbers of vaccines received) in the figures.

Point 19: Section Numbering

 Discussion, Limitations, Conclusions are all "Section 1"

Response 19: The Reviewer has a good point. We reformulated Discussion as Section 4, Limitations as Section 5 and Conclusions as Section 6.

Reviewer 2 Report

Thank you for an interesting manuscript investigating the prognosis in patients with cirrhosis after COVID-19 infection as well as patientes with cirrhosis without COVID and patients with COVID wothout cirrhosis, but with other liver problems.

There are some main points in the manuscript, that are unclear, and are improtant to be clarified, to make the results interpretable:

- The inclusion and exclusion criteria are unclear, group C seems (according to Table 1, but not Table 2) to requre vaccination against COVID, but it is unclear, if this is required in all patients (it sounds this way in the text) or  not (according to Table 2 and other places in the text).

- It is unclear how the patients included in the study were chosen. At multiple places the study is described as matched, but it is not made clear how the matching was performed and on which characteristica.

- The comparability of the groups is challenging, with patients with preexisting cirrhosis and COVID compared with patients with cirrhosis, without COVID (but still hospitalized due to some other, unspecified reason) with patients with COVID and liver injury.

Moreover, there are some specific points to consider:

- Line 44: The meaning of the abbrevation CLD should be explained the first time it is used.

- Table 1: It is stated that patients with a antigen test were excluded. Probably, they only were excluded, if they did not also have a PCR test?

-Table 1 (and in the ext): It is stated multiple times but not explained, why patients recieving the Janssen vaccine were excluded.

Line 163: How was the adjustment for vaccination performed? In the Kaplan-Meier figures it looks like stratification.

Line 214: The word "resulting" indicates causality, and in this case there could easily be both selection bias and confounding explaining the association.

Figure 5: It is not clear, what this figure shows, as it shows not characteristics by groups but groups by characteristics.

Section 3.4: How was decided who recieved which vaccine? Different vaccine recommendations for different patient groups could explain all of these differences.

Line 240: It is unclear what the "However" here refers to.

Figure 6: The "Total cases" figure is hard to interprete, as it mixes three very different groups of patients.

Page 15 (no line numbers here): Other vaccines than the mRNA vaccines should be discussed in the discussion.

Informed consent statement: It should made clear, why consent was not required.

Author Response

Response to Reviewer 2 Comments

Comments and Suggestions for Authors

Thank you for an interesting manuscript investigating the prognosis in patients with cirrhosis after COVID-19 infection as well as patientes with cirrhosis without COVID and patients with COVID wothout cirrhosis, but with other liver problems.

There are some main points in the manuscript, that are unclear, and are improtant to be clarified, to make the results interpretable:

Point 1: The inclusion and exclusion criteria are unclear, group C seems (according to Table 1, but not Table 2) to requre vaccination against COVID, but it is unclear, if this is required in all patients (it sounds this way in the text) or  not (according to Table 2 and other places in the text).

Response 1: We fully agree with the Reviewer. We reformulated inclusion and exclusion criteria.  As demonstrated in Table 2, patient groups (A,B,C) included both vaccinated and unvaccinated patients. As demonstrated in Table 1, exclusion criteria was only the single-dose vaccine administration.  Therefore, GROUP C did not require vaccination as inclusion criteria.

Point 2: It is unclear how the patients included in the study were chosen. At multiple places the study is described as matched, but it is not made clear how the matching was performed and on which characteristica.

Response 2: The Reviewer has an important point. We clarified how the matching was performed and on which characteristica as follows:

“Thus, GROUP B patients were matched with 54 cirrhosis patients without COVID-19 infection (GROUP C) in respect to age and gender in 1:1 ratio (Figure 1). The cirrhosis controls without COVID-19 cohort included all laboratory-confirmed SARS-CoV-2 negative, adult patients who were previously diagnosed with liver cirrhosis and were admitted in the participating centres due to deterioration in cirrhosis stage between March 2020 and May 2022.”

Point 3: The comparability of the groups is challenging, with patients with preexisting cirrhosis and COVID compared with patients with cirrhosis, without COVID (but still hospitalized due to some other, unspecified reason) with patients with COVID and liver injury.

Response 3: The Reviewer has a good point. We compared COVID-19 patients with elevated liver enzymes on admission according to the presence of cirrhosis (GROUP A and GROUP B). Thus, COVID-19 patients with cirrhosis were matched with cirrhosis controls without COVID-19 to investigate the impact of COVID-19 on hospital outcomes. As mentioned in Line 84, GROUP C was hospitalised due to deterioration in cirrhosis stage.

Moreover, there are some specific points to consider:

Point 4: Line 44: The meaning of the abbrevation CLD should be explained the first time it is used.

Response 4: We fully agree with the Reviewer. We rephrased the sentence as follows:

“Furthermore, patients with decompensated cirrhosis following SARS-CoV-2 infection were more susceptible to Intensive Care Unit (ICU) admission, renal replacement therapy and invasive mechanical ventilation compared to those without chronic liver disease (CLD) [6].”

Line 73: „Patients with CLD were identified as cirrhosis cases.”

Point 5: Table 1: It is stated that patients with a antigen test were excluded. Probably, they only were excluded, if they did not also have a PCR test?

Response 6: The Reviewer has a good point. As demonstrated in Table 1, antigen test-confirmed SARS-CoV-2 infection without PCR positivity was excluded from our study as follows:

„Rapid Antigen Test-confirmed SARS-CoV-2 infection without PCR-positivity”

Point 6: Table 1 (and in the ext): It is stated multiple times but not explained, why patients recieving the Janssen vaccine were excluded.

Response 6: The Reviewer has a good point. We explained it in Line 147-149 as follows:

„Individuals receiving only one dose of COVID-19 vaccine including the single-dose Janssen vaccine were excluded from our analysis due to inadequate primary immunisation.”

Point 7: Line 163: How was the adjustment for vaccination performed? In the Kaplan-Meier figures it looks like stratification.

We fully agree with the Reviewer. We rephrased it as follows:

“In addition, survival probabilities of three groups stratified by primary vaccination with different COVID-19 vaccines were displayed on a Kaplan-Meier plot and compared with a log-rank test.”

Point 8: Line 214: The word "resulting" indicates causality, and in this case there could easily be both selection bias and confounding explaining the association.

Response 8: The Reviewer has a good point. We rephrased it as follows:

“The prevalence of life-threatening ACLF in GROUP C was higher (14.8%) in comparison with GROUP B (11.5%), which may have contributed to higher in-hospital mortality in GROUP C (Figure 4).”

Point 9: Figure 5: It is not clear, what this figure shows, as it shows not characteristics by groups but groups by characteristics.

Response 9: The Reviewer has a good point. We reformulated the legend as follows:

“Proportion of different Child-Turcette-Pugh (CTP) stages in all and deceased patients with cirrhosis.”

Point 10: Section 3.4: How was decided who recieved which vaccine? Different vaccine recommendations for different patient groups could explain all of these differences.

Response 10: The Reviewer has a good point. We reformulated it as follows:

“In Hungary, Pfizer-BioNTech was the first vaccine to be used in cirrhosis. Primary vaccination campaign using mRNA vaccines (Pfizer-BioNTech and Moderna) started on 15 January 2021. During the third wave in March and April 2021, five different vaccines were widely used in cirrhosis: two mRNA-based vaccines (BNT162b2-Pfizer-BioNTech and mRNA-1273-Moderna, two vector vaccines (AZD1222-Astra Zeneca, Gam-COVID-Vac-Sputnik V) and one inactivated vaccine (HB02-Sinopharm) [14].”

Point 11: Line 240: It is unclear what the "However" here refers to.

Response 11: : The Reviewer has a good point. We rephrased it as follows:

“In addition, patients in GROUP B were significantly associated with O2 support and mechanical ventilation due to the onset of COVID-19 induced respiratory failure.”

Point 12: Figure 6: The "Total cases" figure is hard to interprete, as it mixes three very different groups of patients.

Response 12: The Reviewer has a good point. Figure 6 „Total cases „ have been excluded from our study.

Point 13: Page 15 (no line numbers here): Other vaccines than the mRNA vaccines should be discussed in the discussion.

Response 13: The Reviewer has an important point. We reformulated discussion as follows:

„Apart from mRNA vaccines, the Hungarian vaccination campaign included AZD1222-AstraZeneca, Gam-COVID-Vac-Sputnik V and HB02-Sinopharm vaccines. In a nationwide, retrospective study investigating 3 740 066 Hungarian citizens, the overall estimated effectiveness of AstraZeneca, Sputnik-V and Sinopharm against COVID-19 related death was lower compared to Pfizer-BioNTech and Moderna [14]. Our study also found evidence that primary vaccination with viral vector and inactivated vaccines was less effective at preventing in-hospital mortality in COVID-19 patients, regardless of liver cirrhosis.”

Point 14: Informed consent statement: It should made clear, why consent was not required.

Response 14: The Reviewer has a good point. We rephrased it as follows:

“Informed consent was not needed owing to the retrospective design”

Reviewer 3 Report

The manuscript submitted by the authors may be of potential interest to a wide audience of Vaccines readers and the illustrated dataset is interesting. Nevertheless, the statistical analysis should be improved, with particular reference to the quantification of uncertainties of the achieved estimates. Therefore I suggest that the article be subject to major revision before publication, taking into account the following comments.

- As the cases were recorded over a prolonged time interval (March 2020 - May 2022), a possible bias is expected as individuals belonging to different cohorts were exposed to different contagion conditions, due both to the fact that in this time interval different variants of the virus have spread, both at a different contagion level of the population. This limitation of sample quality should be emphasized in the "Limitation" section and in the conclusions.

- The statistical sample is in some cases very small. By way of example, mortality rates of 11.8% in Group C (7 deaths among 54 cases) and 9.6% in Group B (5 deaths among 52 cases) are highlighted in the Abstract. Nevertheless, assuming a binomial distribution, the respective 90% confidence intervals are [5.1% - 19.1%] in Group C and [7.6% - 23%] in Group B. Therefore these intervals largely overlap and so it would certainly be more appropriate to talk about non-significant differences in the two recorded mortality rates. Another example of statistics based on inconsistent samples is given by the diagrams in Figs. 6B and 6D. Authors should provide estimates of the uncertainty of the obtained statistical parameters (e.g. by means of confidence intervals) and clarify to the reader which results are statistically significant and which are not.

Minor Comments

- Section Introduction, line 44: At the first mention of an acronym, even when its use is widespread, it should be written in full, followed by the acronym in parentheses. I suggest writing the meaning of CLD at the first occurrence.

Author Response

Response to Reviewer 3 Comments

The manuscript submitted by the authors may be of potential interest to a wide audience of Vaccines readers and the illustrated dataset is interesting. Nevertheless, the statistical analysis should be improved, with particular reference to the quantification of uncertainties of the achieved estimates. Therefore I suggest that the article be subject to major revision before publication, taking into account the following comments.

Point 1:  As the cases were recorded over a prolonged time interval (March 2020 - May 2022), a possible bias is expected as individuals belonging to different cohorts were exposed to different contagion conditions, due both to the fact that in this time interval different variants of the virus have spread, both at a different contagion level of the population. This limitation of sample quality should be emphasized in the "Limitation" section and in the conclusions.

Response 1: We fully agree with the Reviewer. Section „Limitation” needed reformulating as follows:

“As the cases were recorded over a prolonged time interval, a potential bias is expected as different cohorts were exposed to different contagion conditions. The study period overlapped with different waves of multiple variants of SARS-CoV-2, which may have influenced the effectiveness of COVID-19 vaccines investigated in our analysis”.

Point 2: The statistical sample is in some cases very small. By way of example, mortality rates of 11.8% in Group C (7 deaths among 54 cases) and 9.6% in Group B (5 deaths among 52 cases) are highlighted in the Abstract. Nevertheless, assuming a binomial distribution, the respective 90% confidence intervals are [5.1% - 19.1%] in Group C and [7.6% - 23%] in Group B. Therefore these intervals largely overlap and so it would certainly be more appropriate to talk about non-significant differences in the two recorded mortality rates. Another example of statistics based on inconsistent samples is given by the diagrams in Figs. 6B and 6D. Authors should provide estimates of the uncertainty of the obtained statistical parameters (e.g. by means of confidence intervals) and clarify to the reader which results are statistically significant and which are not.

 Response 2: We fully agree with the Reviewer. In the abstract, we reformulated mortality rates as follows:

“Of cirrhosis cases, we found no difference (p=0.761) in mortality rates in controls without COVID-19 (11.8%) compared to those with COVID-19 (9.6%).”

In Figure 6, we included estimates with confidence intervals, where appropriate.

Minor Comments

Point 3: Section Introduction, line 44: At the first mention of an acronym, even when its use is widespread, it should be written in full, followed by the acronym in parentheses. I suggest writing the meaning of CLD at the first occurrence.

Response 3: We fully agree with the Reviewer. We reformulated it as follows:

Furthermore, patients with decompensated cirrhosis following SARS-CoV-2 infection were more susceptible to Intensive Care Unit (ICU) admission, renal replacement therapy and invasive mechanical ventilation compared to those without chronic liver disease (CLD) [6].

Round 2

Reviewer 1 Report

Thank you for the substantial improvements to your paper. It reads much better now. I believe it will be of interest to the wider community.

I have a few minor comments on the new version:

ABSTRACT

_________________

p=0.761

just write

p=0.76

to avoid false precision

_________________

"Outstandingly primary vaccination with BNT162b2 vaccine was the most effective in preventing acute hepatic decompensating events, COVID-19 infection requiring hospital admission and in-hospital mortality."

If you put a sentence in the Abstract, it is outstanding. So no need to write it.

just write

"Primary vaccination with BNT162b2 vaccine was the most effective in preventing acute hepatic decompensating events, COVID-19 infection requiring hospital admission and in-hospital mortality."

_________________

INTRODUCTION

_________________

"Despite being respiratory failure the leading cause (71%) in COVID-19,"

still needs clarification. Cause of what?

Maybe you mean

"Despite respiratory failure being the leading cause (71%) of death in COVID-19,"

or perhaps

"Despite respiratory failure being the leading complication (71%) in COVID-19,"

____

Figure 1

maybe explain a bit better. If you match 1:1 then the numbers have to be EQUAL. 52 does not equal 54. So it is not "1:1"

___

Table 2 & Table 3

don't use more than two significant figures for reporting p-values

the extra significant figures do not add value and they clutter the table.

Author Response

Response to Reviewer 1 Comments

Thank you for the substantial improvements to your paper. It reads much better now. I believe it will be of interest to the wider community.

I have a few minor comments on the new version:

Point 1: ABSTRACT

p=0.761

just write

p=0.76

to avoid false precision

Response 1: The Reviewer has a good point. We reformulated the p-value in the abstract to avoid false precision as follows:

„p=0.76”

________________

Point 2: "Outstandingly primary vaccination with BNT162b2 vaccine was the most effective in preventing acute hepatic decompensating events, COVID-19 infection requiring hospital admission and in-hospital mortality."

If you put a sentence in the Abstract, it is outstanding. So no need to write it.

just write

"Primary vaccination with BNT162b2 vaccine was the most effective in preventing acute hepatic decompensating events, COVID-19 infection requiring hospital admission and in-hospital mortality."

Response 2: We fully agree with the Reviewer. We rephrased the sentence in the abstract as follows:

"Primary vaccination with BNT162b2 vaccine was the most effective in preventing acute hepatic decompensating events, COVID-19 infection requiring hospital admission and in-hospital mortality."

_________________

Point 3: INTRODUCTION

"Despite being respiratory failure the leading cause (71%) in COVID-19,"

still needs clarification. Cause of what?

Maybe you mean

"Despite respiratory failure being the leading cause (71%) of death in COVID-19,"

or perhaps

"Despite respiratory failure being the leading complication (71%) in COVID-19,"

Response 3: The Reviewer has a good point. The sentence needs to be clarified accordingly:

"Despite respiratory failure being the leading cause (71%) of death in COVID-19,"

_________________

Point 4: Figure 1

maybe explain a bit better. If you match 1:1 then the numbers have to be EQUAL. 52 does not equal 54. So it is not "1:1"

Response 4: The Reviewer has a good point. We explained it a bit better in Line 81-83 as follows:

“The matching ratio was very close to 1:1. The representation of patients in cirrhosis groups was proportional.”

In addition, we excluded the 1:1 matching ratio from Figure 1.

Point 5: Table 2 & Table 3

don't use more than two significant figures for reporting p-values

the extra significant figures do not add value and they clutter the table.

Response 5: We fully agree with the Reviewer. We reformulated Table 2 and Table 3 to report significant p-values as p<.001 and p<.05. Furthermore, extra significant figures were excluded from our study. Therefore, p=0.024 was reformulated in Table 3 as follows:

„p<0.05”

Reviewer 2 Report

The authors have made relevant chsanges to the manuscript, taking into account all my points.
